# Mitochondrial, lysosomal and DNA damages induced by acrylamide attenuate by ellagic acid in human lymphocyte

**Ahmad Salimi**[1,2]*, **Elahe Baghal**[1,3], **Hassan Ghobadi**[4], **Niloufar Hashemidanesh**[1,3], **Farzad Khodaparast**[1,3], **Enayatollah Seydi**[5,6]*

**1** Department of Pharmacology and Toxicology, School of Pharmacy, Ardabil University of Medical Sciences, Ardabil, Iran, **2** Traditional Medicine and Hydrotherapy Research Center, Ardabil University of Medical Sciences, Ardabil, Iran, **3** Students Research Committee, School of Pharmacy, Ardabil University of Medical Sciences, Ardabil, Iran, **4** Faculty of Medicine, Internal Medicine Department (Pulmonary Division), Ardabil University of Medical Sciences, Ardabil, Iran, **5** Department of Occupational Health and Safety Engineering, School of Health, Alborz University of Medical Sciences, Karaj, Iran, **6** Research Center for Health, Safety and Environment, Alborz University of Medical Sciences, Karaj, Iran

* salimikd@yahoo.com, a.salimi@pharmacy.arums.ac.ir(AS); enayat.seydi@yahoo.com(ES)

**Data Availability Statement:** All relevant data are within the manuscript and its Supporting Information files.

## Abstract

Acrylamide (AA), is an important contaminant formed during food processing under high temperature. Due to its potential neurotoxicity, reproductive toxicity, hepatotoxicity, immunotoxicity, genotoxicity and carcinogenicity effects, this food contaminant has been recognized as a human health concern. Previous studies showed that acrylamide-induced toxicity is associated with active metabolite of acrylamide by cytochrome P450 enzyme, oxidative stress, mitochondrial dysfunction and DNA damage. In the current study, we investigated the role of oxidative stress in acrylamide's genotoxicity and therapeutic potential role of ellagic acid (EA) in human lymphocytes. Human lymphocytes were simultaneously treated with different concentrations of EA (10, 25 and 50 μM) and acrylamide (50 μM) for 4 h at 37°C. After 4 hours of incubation, the toxicity parameters such cytotoxicity, ROS formation, oxidized/reduced glutathione (GSH/GSSG) content, malondialdehyde (MDA) level, lysosomal membrane integrity, mitochondria membrane potential (ΔΨm) collapse and 8-hydroxy-2'-deoxyguanosine (8-OHdG) were analyzed using biochemical and flow cytometry evaluations. It has been found that acrylamide (50 μM) significantly increased cytotoxicity, ROS formation, GSH oxidation, lipid peroxidation, MMP collapse, lysosomal and DNA damage in human lymphocytes. On the other hand, cotreatment with EA (25 and 50 μM) inhibited AA-induced oxidative stress which subsequently led to decreasing of the cytotoxicity, GSH oxidation, lipid peroxidation, MMP collapse, lysosomal and DNA damage. Together, these results suggest that probably the co-exposure of EA with foods containing acrylamide could decrease mitochondrial, lysosomal and DNA damages, and oxidative stress induced by acrylamide in human body.

**Funding:** Ahmad Salimi This study was supported by Ardabil of Medical Sciences, Deputy of Research with ethics code IR.ARUMS. REC.1398.496.

**Competing interests:** The authors declare that they have no conflict of interest.

## Introduction

One of heat-induced toxic substances in during food processing techniques is acrylamide (AA) that has recently received much scientific interest [1]. In the past, acrylamide was found in many industrial processes, such as in the manufacture of paper, glues, plastics, in the treatment of drinking water and wastewater and as a component in the cigarette smoke [2]. For the first time in April 2002, acrylamide was also reported by Swedish researchers in consumer products, particularly starchy foods such as grain products and potato [3]. The Stockholm University and Swedish National Food Administration (SNFA) researchers detected high levels (150–4,000 mg/kg) and moderate levels (5–50 mg/kg) of acrylamide in heated carbohydrate-rich and protein foods, respectively [4]. Exposure humans to acrylamide can be occurred through inhalational, dermal, and oral routes [1]. Due to wide range of acrylamide exposure sources diet, drinking water, smoking, and environmental/occupational sources, many people can be exposed to this chemical through oral, dermal or inhalation [1,5,6]. After exposure, it is showed that acrylamide is completely and rapidly absorbed and distributed to the peripheral tissues in animals and humans [7]. A study in healthy volunteers have showed that acrylamide is able to reach any human tissue. In the body acrylamide is metabolized to glycidamide as active metabolite by the cytochrome P450 enzyme (CYP450), where is known to be more reactive toward macromolecules such proteins, lipids and DNA than the parent acrylamide compound [8]. Acrylamide is associated with a wide range of different toxicities in human and animal such as neurotoxicity, reproductive toxicity, hepatotoxicity, immunotoxicity, genotoxicity and carcinogenicity [9]. Also, this substance has been categorized as a probable human carcinogen by the International Agency for Research on Cancer (IARC) [10]. Due to definite exposure of humans to acrylamide for various sources, probably detoxification of this compound inside the body can be a suitable approach to reduce its toxicity. It is well confirmed that oxidative stress and mitochondrial dysfunction have a main role in acrylamide-induced toxicity [11]. Therefore, natural compounds with antioxidant and antitumorigenic properties can probably play a promising role in inhibiting the toxicity caused by acrylamide.

Natural products extracted from plants have been demonstrated to reverse acrylamide-induced toxicity [9]. For example, chrysin as plant-derived natural product, reduced acrylamide-induced neurotoxicity in Wistar rats due to its high antioxidant properties [12]. Also, quercetin is able to protect against acrylamide-induced neurotoxicity [13]. Berry juices significantly enhance the growth of acrylamide-exposed yeast cells, and decreased the level of ROS [14]. Ellagic acid (EA) as a plant-derived natural product found in pomegranate (Punica granatum) and the family of ellagitannins (ETs) has different pharmacological effects [15]. Ellagic acid has generated a noticeable scientific interest because of beneficial health effects against many oxidative-linked diseases, including neurodegenerative and cancer diseases [15]. It has been reported that ellagic acid can prevent of oxidative stress, inflammation, and histopathological alterations in acrylamide-induced hepatotoxicity in wistar rats [16]. Also, et al at 2019 reported that ellagic acid has neuroprotective effects against acrylamide-induced neurotoxicity in rats [17]. Due to the promising effect of ellagic acid on the oxidative stress, genotoxicity and cytotoxicity, it deserves to be more studded. Therefore, in the current study for the first time we evaluate the effect of ellagic acid against acrylamide-induced toxicity in primary lymphocytes obtained from humans.

## Materials and methods

### Chemicals

MTT 3-(4,5-dimethylthiazol-2-yl)-2,5-diphenyltetrazolium bromide, 2′,7′-dichlorofuorescin diacetate (DCFH-DA), Rhodamine123, Bovine Serum Albumin (BSA), Trypan blue, Acridine

Orange, N-(2-hydroxyethyl) piperazine-N′-(2-ethanesulfonic acid) (HEPES), Acrylamide with CAS-Nummer: 79-06-1 (A9099), Ellagic Acid, Fetal Bovine Serum (FBS), 2-mercaptoethanol and Antibiotic-Antimycotic Solution were obtained from the Sigma Chemical Co. RPMI1640 was purchased from GIBCO (USA). Ficoll-paque PLUS was provided from Ge Healthcare Bio-Science Company.

## Sample collection and ethic statement

The study was performed by using isolated human lymphocytes as primary cells from 10 healthy, young (20–24 years) subjects (four men and six women). The criteria of acceptability to ensure reliability of the experiment were: receiving any medical therapy, from non-smoker and non-alcoholic persons, without serious illness and having good health. The choice of healthy volunteers was performed at the Internal Medicine Department (Pulmonary Division), Faculty of Medicine, Ardabil University of Medical Sciences, under the guidance of an expert physician (Hassan Ghobadi). Before blood sampling, the informed consent form was signed by the participants. This project was performed in Ardabil University of Medical Sciences at school of pharmacy. The dedicated approved ethical code by research ethic committee of Ardabil University of Medical Sciences for this study is IR.ARUMS.REC.1398.496.

## Lymphocytes isolation

Peripheral blood obtained from young donors were used for isolation of lymphocytes. Briefly, 5 ml of blood containing anticoagulant was gently mixed with 5 ml of normal saline in a 15 ml sterile tube. Then 5 ml of diluted blood was added to two sterile tube contains 3 ml of Ficoll-paque PLUS and centrifuged at 2500 rpm at 4 C˚ for 20 min. After centrifugation, the buffy coat was isolated and added to a new sterile falcon tube. The buffy coat was centrifuged at 1500 rpm at 4 C˚ for 10 min, and the pellet was suspended in erythrocyte lysis buffer and incubated at 37˚C for 5 min. After centrifugation, the supernatant was discarded, and the pellet were washed with RPMI1640 supplemented with 10% fetal bovine serum (FBS) and l-glutamine and centrifuged at 2000$g$ for 7 min. The isolated cells were resuspended in RPMI1640 medium supplemented with L-glutamine and 10% FBS at 37˚C in normal condition with a humidified atmosphere and 5% $CO_2$. Before the experiments, the number of live cells was measured by Trypan Blue and the cell viability was more than 95%. The lymphocyte density used in the tests was $10 \times 10^6$ cells/ml [18].

## Treatment of isolated lymphocytes

For all experiments isolated lymphocytes were cultured in RPMI1640 with supplemented l-glutamine and 10% FBS at 37˚C with 5% $CO_2$ in a humidified atmosphere. After the initial period of incubation, isolated lymphocytes were treated with dimethyl sulfoxide (DMSO) (.05%) as control group, $IC_{50}$ 4h acrylamide, $IC_{50}$ 4h acrylamide + 10μM EA, $IC_{50}$ 4h acrylamide + 25μM EA, $IC_{50}$ 4h acrylamide + 50μM EA and 50μM EA at 37˚C. Each experiment included a positive control (data not shown).

## Measurement of cell viability

Briefly, isolated lymphocytes ($10^4$ cells per well) were incubated in 96-well culture plates in 100 μl of RPMI1640 supplemented with l-glutamine and 10% FBS at 37˚C with 5% $CO_2$ in a humidified atmosphere for 4 hours according to the above grouping. After treatment 25 μl of MTT (0.5 mg/ml) was added to each well. After the 4 h incubation at 37˚C, 100 ml of DMSO was added to dissolve the water-insoluble formazan salt. Then, the absorbance at λ = 570 nm

was measured with an ELISA microplate reader. The viability was represented as the percentage absorbance compared with untreated control group. Each experiment was performed three times [19].

## Measurement of reactive oxygen species

The intracellular ROS formation was measured using 2′,7′-dichlorofuorescin diacetate (DCFH-DA). Inside the cells DCFH-DA hydrolyzed to the nonfluorescent DCFH by using intracellular esterase, which in the presence of ROS, could be rapidly oxidized to the highly fluorescent 2,7-dichlorofluorescein (DCF). After treatment for 4 hours according to the above grouping, the isolated lymphocytes were washed with PBS and incubated with DCFH-DA at a final concentration of 5 μM for an additional 20 min at 37°C in the dark. The isolated lymphocytes that were previously stained with DCFH-DA were separated from the medium by 1 min centrifugation at 1000 rpm. To remove the additional DCFH-DA from the medium the cell washing was performed twice using PBS. The fluorescent intensity of the cell suspensions was measured by flow cytometry (Cyflow Space-Partec, Germany) and mean of fluorescence intensities were analyzed by software (FlowJo) [20]. The results were expressed as fluorescent intensity per $10^4$ cells

## Measurement of mitochondrial membrane potential collapse

To monitor the membrane potential of mitochondria we used rhodamine 123 as a fluorescent probe. This fluorescent probe is lipophilic cation accumulated by mitochondria in proportion to $\Delta\Psi$. Upon accumulation, rhodamine 123 exhibit a red shift in both its absorption and fluorescence emission spectra. The fluorescence intensity is quenched when the dye is accumulated by mitochondria. Briefly, after treatment for 4 hours according to the above grouping, the isolated lymphocytes were washed with PBS and incubated with rhodamine 123 at a final concentration of 5 μM for an additional 20 min at 37°C in the dark. The isolated lymphocytes that were previously stained with rhodamine 123, were separated from the medium by 1 min centrifugation at 1000 rpm. To remove the additional rhodamine 123 from the medium the cell washing was performed twice using PBS. The fluorescent intensity of the cell suspensions was measured by flow cytometry (Cyflow Space-Partec, Germany) and mean of fluorescence intensities were analyzed by software (FlowJo). The results were expressed as fluorescent intensity per $10^4$ cells [21].

## Measurement of lysosomal membrane destabilization

Acridine orange was used to measure lysosomal membrane destabilization as described previously [22]. Acridine orange, a lysosomotropic weak base, accumulates in lysosome on the basis of proton trapping and emits red fluorescence in intact lysosomes with high concentrations, acridine orange and emits green fluorescence in the cytosol and the nucleus with low concentrations [23]. Therefore, lysosomal membrane destabilization can be determined by detecting changes in either green fluorescence or red fluorescence. Briefly, after treatment for 4 hours according to the above grouping, the isolated lymphocytes were washed with PBS and incubated with acridine orange at a final concentration of 5 μM for an additional 20 min at 37°C in the dark. The isolated lymphocytes that were previously stained with acridine orange were separated from the medium by 1 min centrifugation at 1000 rpm. To remove the additional acridine orange from the medium the cell washing was performed twice using PBS. The fluorescent intensity of the cell suspensions was measured by flow cytometry (Cyflow Space-Partec, Germany) and mean of fluorescence intensities were analyzed by software (FlowJo). The results were expressed as fluorescent intensity per $10^4$ cells [21].

## Measurement of lipid peroxidation

The content of malondialdehyde (MDA) as by-product of lipid peroxidation, was determined by the thiobarbituric acid (TBA) reactive substances during an acid-heating reaction. The MDA-TBA adduct can be easily quantified calorimetrically at 532 nm. Briefly, after treatment for 4 hours according to the above grouping, the isolated lymphocytes were washed with PBS, and then mechanically lysed by homogenizer in a tube containing 1 ml 0.1% (w/v) trichloro-acetic acid (TCA). The micro tubes containing homogenized samples were centrifuged at 10,000 x g for 10 min and the supernatant transferred to new tube containing 4 ml of TBA reagent (containing 20% TCA and 0.5% TBA). Then all the micro tubes were placed in a boiling water bath for 15 min and after quick cooling on ice, the mixture was centrifuged again at $1000 \times g$ for 10 min. The content of MDA in each tube was measured by the absorbance of the supernatant at 532 nm with an ELISA reader [24].

## Measurement of GSH and GSSG contents

GSH and GSSG levels in treated isolated lymphocytes were evaluated using 5, 5'-dithiobis-2-nitrobenzoic acid (DTNB) as the indicator and spectrophotometer method for measurement of GSH and GSSG contents [25]. Briefly, after treatment for 4 hours according to the above grouping, the isolated lymphocytes were washed with PBS twice and mechanically lysed by homogenizer resuspended in 1 mL of 0.1 mol/L phosphate buffer (pH 7.4). The lysate was centrifuged at $8,000 \times g$ at 4˚C for 10 min and supernatant was collected. For assessment of GSH, 100 μl supernatant was mixed with 3 ml 500 mM TRIS–HCl (pH 8.0) buffer containing 10 mM DTNB and incubated at 25˚C for 15 min and for assessment of GSSG, 100 μl of supernatant was added to 3 ml of reaction solution containing glutathione reductase (1 U for each 3 ml reaction solution), 500 mM TRIS–HCl (pH 8.0) buffer, 150 μM NADPH, 1 mM EDTA, 3 mM MgCl2, then was add 100 mM DTNB to a final concentration of 10 mM and incubate at room temperature for 15 min. The produced yellow color was read at 412 nm on a spectrophotometer. GSH and GSSG contents were observed as μM [25].

## Determination of 8-hydroxy-2'-deoxyguanosine (8-OHdG)

Briefly, after treatment for 4 hours according to the above grouping, the cellular DNA was isolated using a DNA extraction kit (iNtRON Biotechnology Inc., Sungnam, Republic of Korea), following the manufacturer's protocol, and quantified. The quantity of 8-OHdG, was determined by kit (OXIS Health Products Inc., Portland, OR, USA) following the manufacturer's protocol and by a calculation on a standard curve measured at 450 nm absorbance using a microplate reader.

## Statistical analysis

Results are presented as mean ± standard deviation. All statistical analyses were performed using the GraphPad Prism version 5. Statistical significance was determined using the one-way ANOVA test, followed by the post-hoc Tukey test. Statistical significance was set at $P < 0.05$. Also, the flow cytometric data was obtained with Cyflow Space-Partec and analyzed by FlowJo software. All experiments were carried-out in triplicate.

# Results

## EA attenuated acrylamide induced cytotoxicity in isolated lymphocytes

The protection of EA on isolated lymphocytes treated with acrylamide was detected by MTT assay, after being treated with EA with different concentrations (10, 25 and 50μM) and

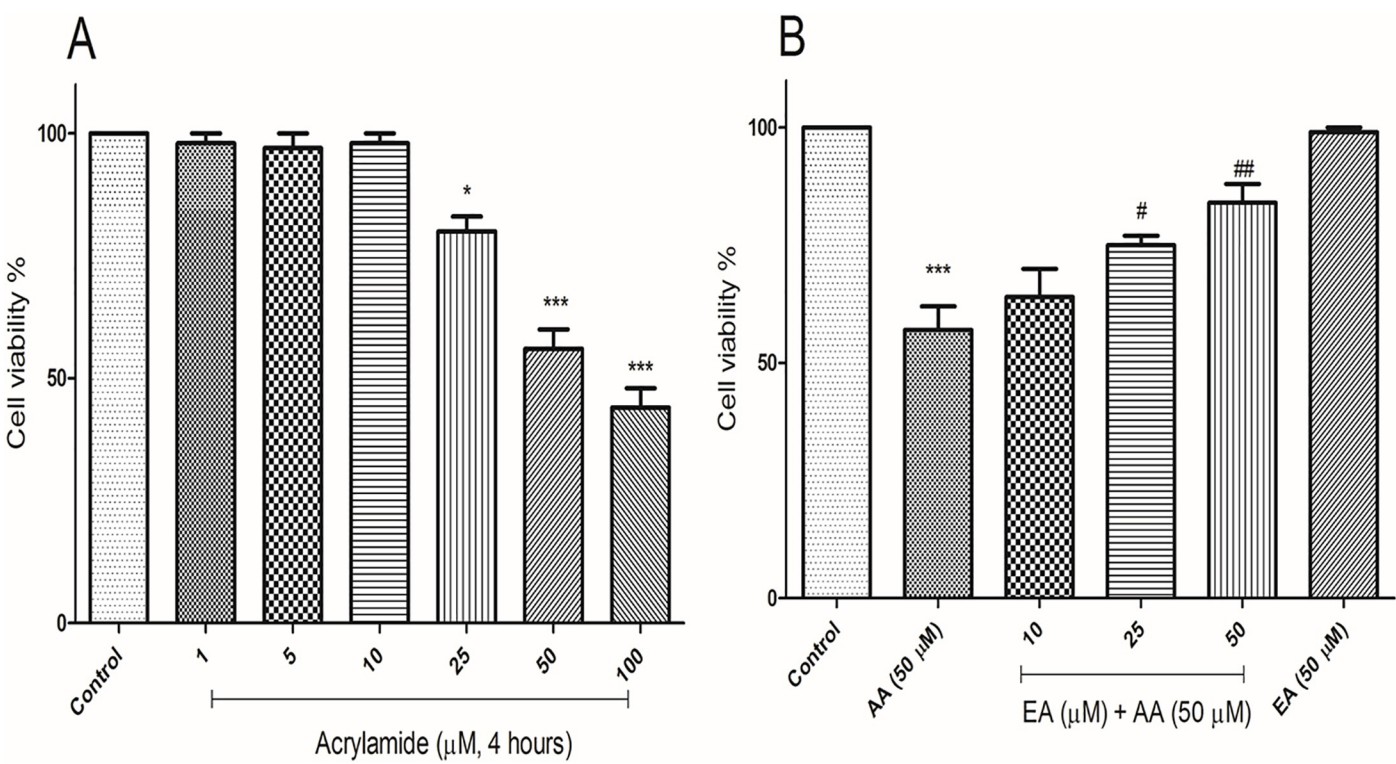

**Fig 1. Ellagic acid reduces cytotoxicity effects of AA in human lymphocytes.** (A) Representative MTT assay of human lymphocytes treated with AA in a range of 1 µM to 100 µM (10,000 cells per well, n = 3 technical replicates; independent experiments repeated at least 3 times) at 4 hours. (B) Representative MTT assay of human lymphocytes treated with AA (50 µM) and EA in a range of 10 µM to 50 µM (10,000 cells per well, n = 3 technical replicates; independent experiments repeated at least 3 times) at 4 hours. Data are presented as mean ± SD, n = 3, ***p < 0.001: Control versus AA; #p < 0.05, ##p < 0.01: AA + EA versus AA-treated human lymphocytes ANOVA, Tukey's test.

acrylamide (50) for 4 h. Compared with the control group, the cell viability in acrylamide (25, 50 and 100 µM) groups was decreased significantly (Fig 1A), but not in the lower concentration groups (1, 5 and 10 µM). Compared with acrylamide (50 µM) group, EA (25 and 50 µM) increased cell viability, especially significant in 50 µM EA group (Fig 1B). The concentrations of EA (10, 25 and 50 µM) were picked for further assay. No significant decrease in the lymphocyte viability was observed when the cells were treated with 50 µM EA alone.

## EA mitigated the ROS formation induced by acrylamide in isolated lymphocytes

To measure the effect of EA on acrylamide-induced oxidative stress in isolated lymphocytes, the level of ROS was measured through DCFH-DA fluorescence by flow cytometry. Results showed that acrylamide (50 µM) increased ROS levels significantly, while EA (25 and 50 µM) achieves anti-oxidative effect by reducing ROS formation (Fig 2A). No significant changes in the lymphocyte ROS formation were observed when the isolated lymphocytes were treated with EA (50 µM) as compared with control group. Buyhylatedhydroxy toluene (BHT) as an antioxidant for confirmation of antioxidant effect of ellagic acid was used (data not shown).

## EA alleviated acrylamide-induced mitochondria injury in isolated lymphocytes

To evaluate the mitochondrial function in isolated lymphocytes, the mitochondrial membrane potential (ΔΨm) was measured by rhodamine 123 staining. As shown in Fig 3, acrylamide

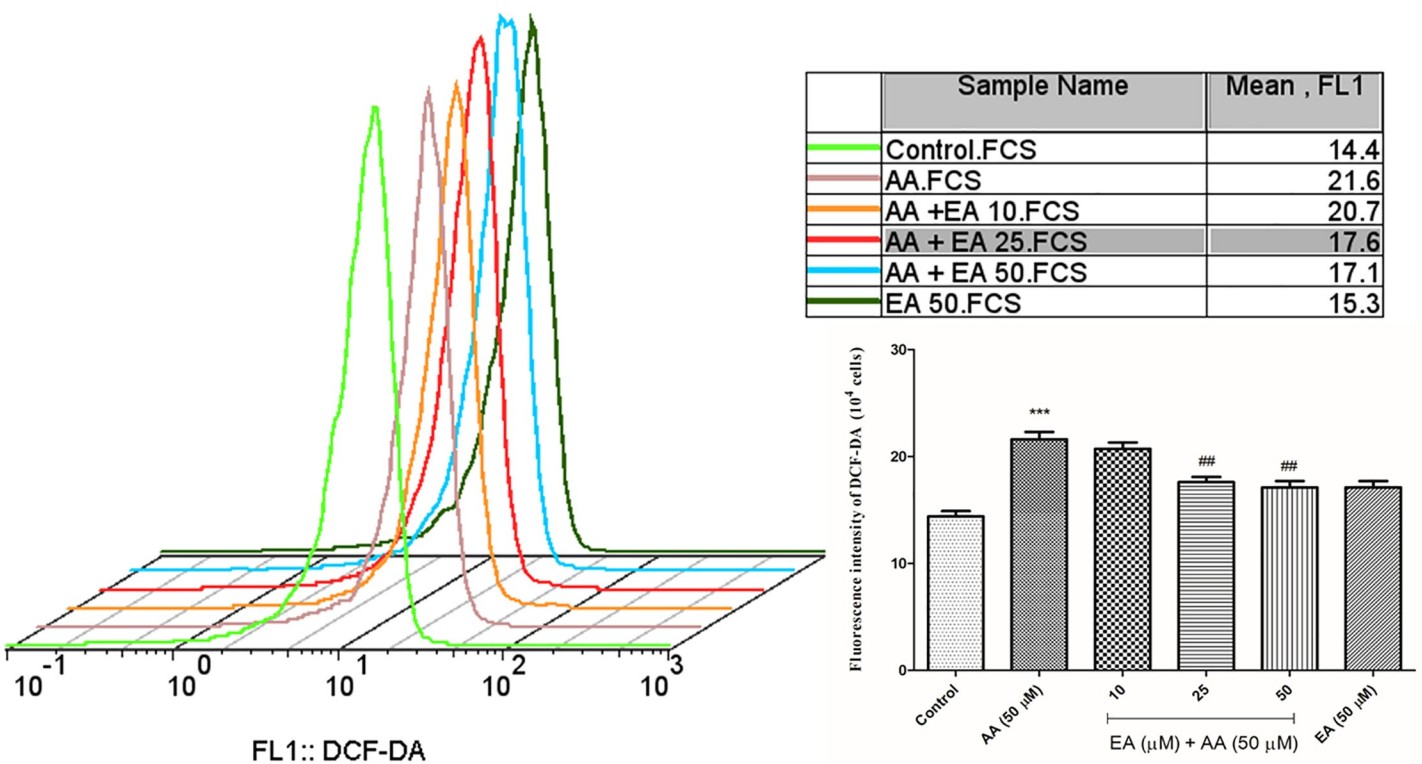

**Fig 2. Ellagic acid decreases ROS formation.** ROS formation was analyzed by means of fluorescence intensity of DCFH-DA following AA exposure (4 hours) in the presence or absence of EA (10, 25 and 50 μM) in human lymphocytes. Bar graphs with statistics. Data are presented as mean ± SD, n = 3, independent experiments repeated at least 3 times, ***p < 0.001: Control versus AA; ##p < 0.01: AA + EA versus AA-treated human lymphocytes ANOVA, Tukey's test.

(50 μM) markedly enhanced fluorescence intensity of rhodamine 123, while EA (25 and 50 μM) treatment partly decreased the fluorescence intensity. It showed that EA significantly alleviated acrylamide-induced mitochondrial membrane potential depolarization and optimized the mitochondrial membrane potential status. Also, no significant alteration in MMP collapse was observed when the isolated lymphocytes were treated with 50 μM EA alone as compared with control group.

## EA alleviated acrylamide-induced lysosomal damages in isolated lymphocytes

To evaluate the lysosomal damages in isolated lymphocytes, the lysosomal membrane destabilization was measured by acridine orange staining. As shown in Fig 4, acrylamide (50 μM) markedly enhanced fluorescence intensity of acridine orange as an indicator of lysosomal damages, while EA (25 and 50 μM) treatment partly decreased the fluorescence intensity of acridine orange. It showed that EA significantly alleviated acrylamide-induced lysosomal damages. Also, no significant alteration was observed when the isolated lymphocytes were treated with 50 μM EA alone as compared with control group.

## EA mitigated acrylamide-induced glutathione depletion in isolated lymphocytes

The levels of GSH and GSSG significantly increased and decreased respectively in the isolated lymphocytes exposed with acrylamide (50 μM). A significant increase in GSH content and a

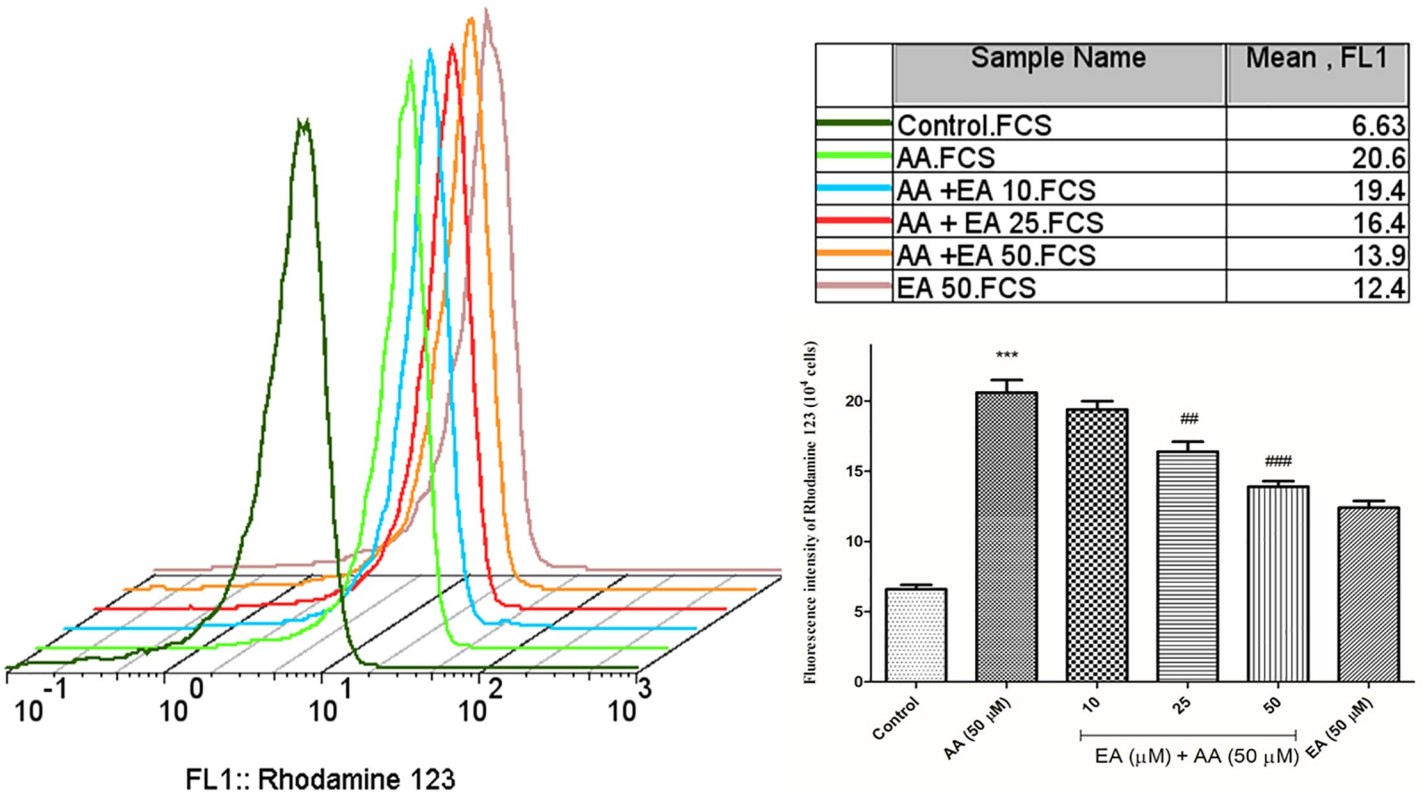

**Fig 3. Ellagic acid decreases mitochondrial damages.** Mitochondrial membrane potential collapse was analyzed by means of fluorescence intensity of rhodamine 123 as a cationic dye following AA exposure (4 hours) in the presence or absence of EA (10, 25 and 50 μM) in human lymphocytes. Bar graphs with statistics. Data are presented as mean ± SD, n = 3, independent experiments repeated at least 3 times, ***p < 0.001: Control versus AA; ##p < 0.01, ###p < 0.001: AA + EA versus AA-treated human lymphocytes ANOVA, Tukey's test.

significant decrease in GSSG was seen when the isolated lymphocytes treated with EA (25 and 50 μM) and acrylamide (50 μM) in comparison with acrylamide group, but did not observe significant different in the levels of GSH and GSSG when compared to the control group. EA (50 μM) alone did not show any change in the isolated lymphocytes GSH and GSSG content when compared with control group (Fig 5).

## EA alleviated acrylamide-induced lipid peroxidation in Isolated lymphocytes

To evaluate the effects of acrylamide on lipid peroxidation, the intracellular levels of MDA were measured in isolated lymphocytes. The concentration of MDA increased from 67 nM in control cells to 102 in 50 μM acrylamide-treated group. However, EA treatment (25 and 50 μM) successfully attenuated the acrylamide-induced increases in intracellular MDA levels in isolated lymphocytes. Also, no significant changes in the intracellular MDA levels were observed when isolated lymphocytes were treated with 50 μM EA alone as compared with control group (Fig 6A).

## The blockade of acrylamide-induced DNA damage by EA in isolated lymphocytes

We subsequently performed 8-hydroxy-2'-deoxyguanosine (8-OHdG) assay to determine whether EA reduces acrylamide-induced DNA damage. Results showed the acrylamide

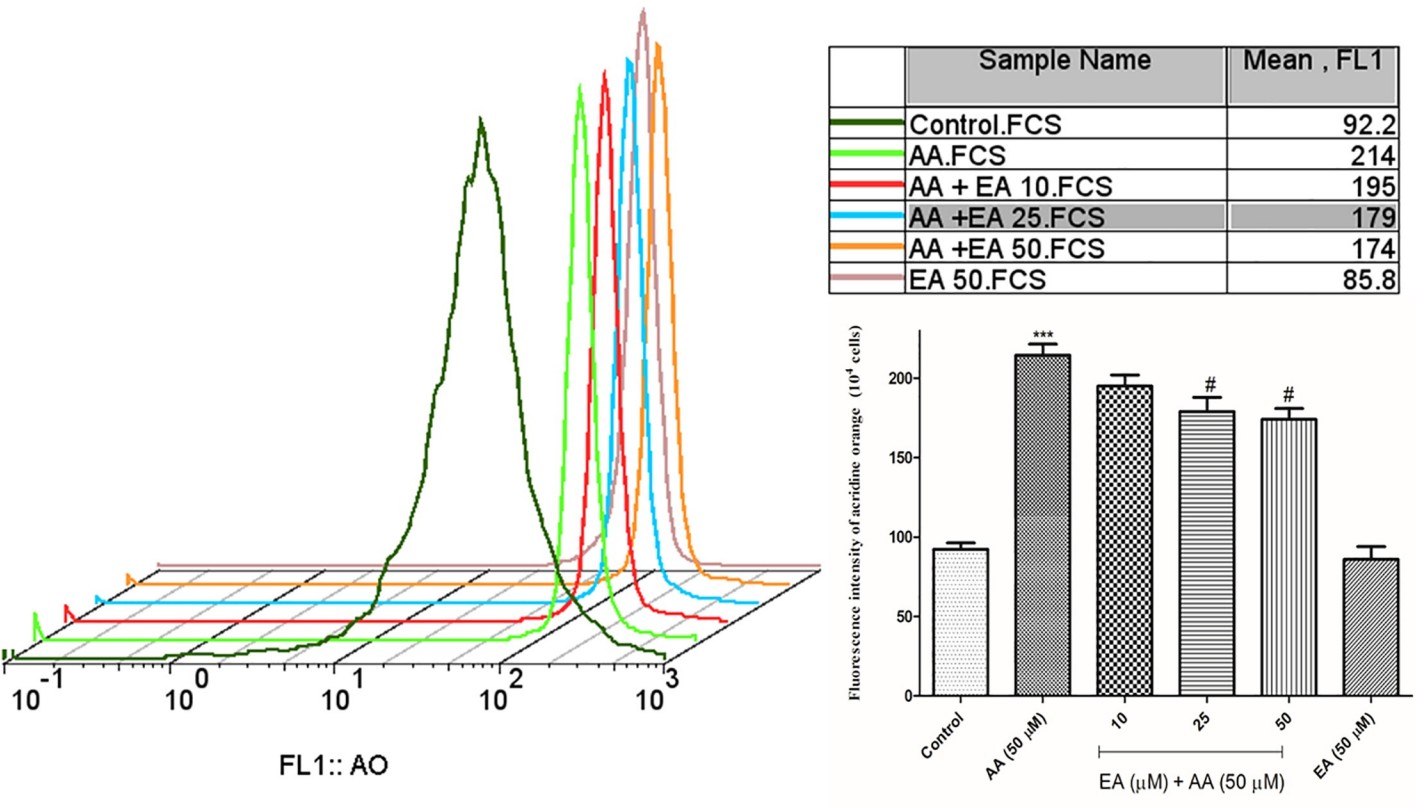

**Fig 4. Ellagic acid decreases lysosomal integrity.** Lysosomal integrity was analyzed by means of fluorescence intensity of acridine orange following AA exposure (4 hours) in the presence or absence of EA (10, 25 and 50 μM) in human lymphocytes. Bar graphs with statistics. Data are presented as mean ± SD, n = 3, independent experiments repeated at least 3 times, ***p < 0.001: Control versus AA; #p < 0.05: AA + EA versus AA-treated human lymphocytes ANOVA, Tukey's test.

(50 μM) treatment significantly increased the production of the 8-OHdG adduct, an oxidative stress-induced DNA damage marker, compared to the control group, but the cotreatment with EA (25 and 50 μM) significantly reduced the production of 8-OHdG by acrylamide. EA (50 μM) alone did not show any change in the production of the 8-OHdG adduct in the isolated lymphocytes when compared with control group (Fig 6B).

## Discussion

In many in vivo and in vitro studies have been reported genotoxicity, cytotoxicity, oxidative stress and mitochondrial dysfunction induced by acrylamide [26,27]. Various genotoxicity experiments showed that acrylamide could develop the genotoxicity and carcinogenicity properties like induction of micronuclei and aberrations in blood cells such as, spleen lymphocytes and peripheral red blood cells, DNA damage in Comet assay in various organs and transgenic gene mutation in liver [28]. Breaks, rearrangements and deletions in DNA, can increase risk of tumor if the DNA damage does not repair or promote into the cell death [28]. Our results showed that acrylamide induce DNA damage in human lymphocytes, these results are consistent with previously published data. Furthermore, several investigations demonstrated that acrylamide induced cancers in various organs including central nervous system, peritesticular mesothelium, mammary gland and thyroid, while the genotoxicity of acrylamide in all of mentioned organs have not been showed [29]. Although the carcinogenic effects of acrylamide have not been proven in many in vitro tests such as comet assay and even at bacterial gene

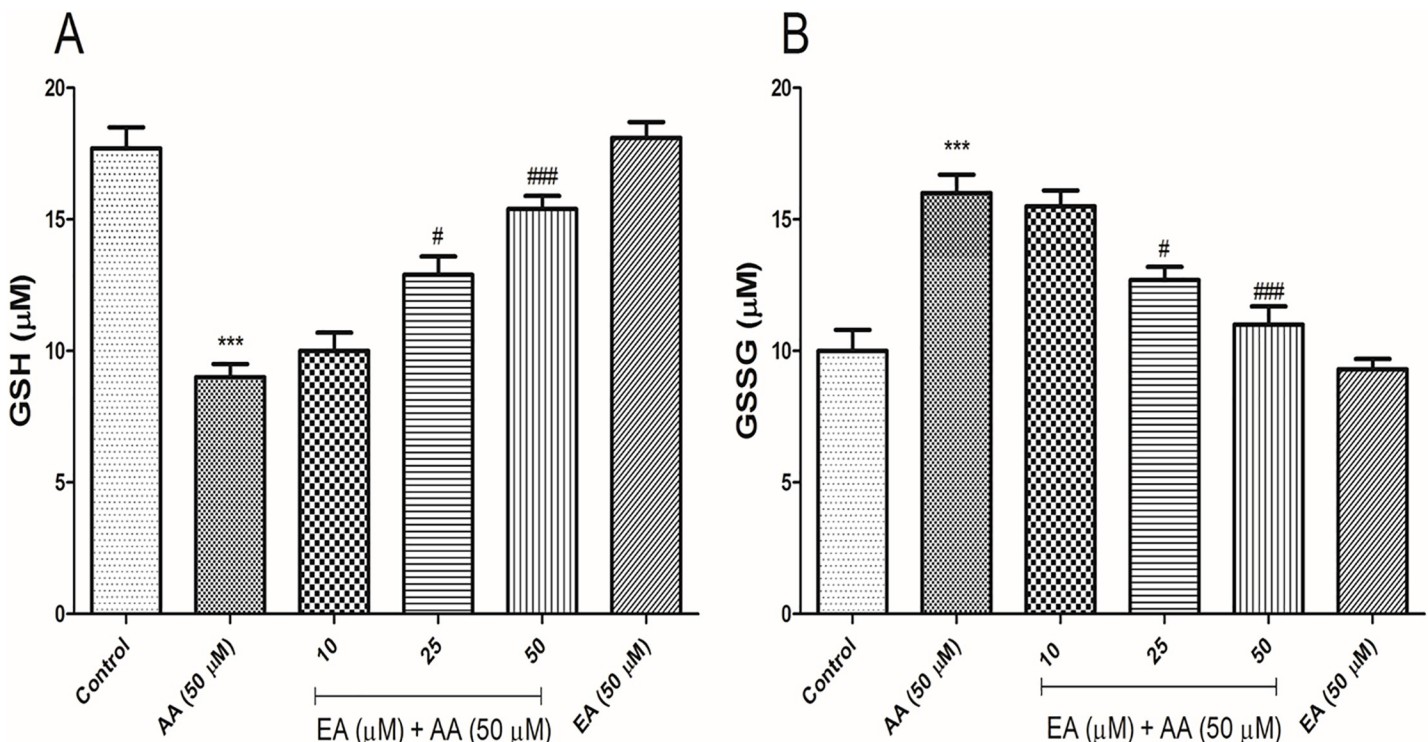

**Fig 5. Addition of ellagic acid decreases AA-induced glutathione depletion in human lymphocytes.** (A) GSH content significantly decreased in AA-treated human lymphocytes while cotreatment EA (50 μM) with AA obviously increased the GSH content in human lymphocytes. (B) Also, AA significantly increased the GSSG content in human lymphocytes after 4 h of exposure while EA (50 μM) significantly decreased the GSSG content in the presence of AA. All experiments were repeated at least three times. $^{***}p < 0.001$: Control versus AA; $\#p < 0.05$, $\#\#\#p < 0.001$: AA + EA versus AA-treated human lymphocytes ANOVA, Tukey's test.

mutation assays high concentrations [30]. Therefore, the use of human cells can be an advantage in identifying the genotoxic effects of acrylamide that has been considered in the current study.

Acrylamide is metabolized to glycidamide as active metabolite by the CYP450 in the body, where is known to be more reactive toward macromolecules such proteins, lipids and DNA than the parent acrylamide compound [31]. Most CYP family members such as CYP1A1, 1A2, P1B1, 2A6, 2B6 and 2E1 are expressed in human lymphocytes. It has been reported that gene expression levels above CYP subfamily were closely correlated within peripheral blood lymphocytes and within the liver [32]. It has been reported that once taken into the body, approximately 50% of acrylamide is metabolized to its epoxide metabolite glycidamide by the cytochrome CYP2E1, which is found in the human lymphocytes [33]. Previous studied have shown the relation between acrylamide-induced toxicity and oxidative stress [34]. Several investigations proved the ability of acrylamide in generation of ROS. ROS can react with the DNA molecule (the deoxyribose backbone and also both the pyrimidine and purine bases) [35]. DNA oxidative damage induced by ROS can play the first step involved in ageing, mutagenesis and carcinogenesis [35]. Our results indicated that acrylamide induce ROS formation in human lymphocytes, these results are consistent with previously published studies. Also, acrylamide-induced ROS generation and its active metabolite can attack to other important macromolecules such proteins and membrane lipids and induce lipid peroxidation and depletion of GSH [36]. Lipid peroxidation as a destructive chain reaction can lead to formation of more free radicals [37]. The adverse effects of free radicals and active metabolite of acrylamide are detoxified by the antioxidant agents such as GSH. GSH is an important intrinsic

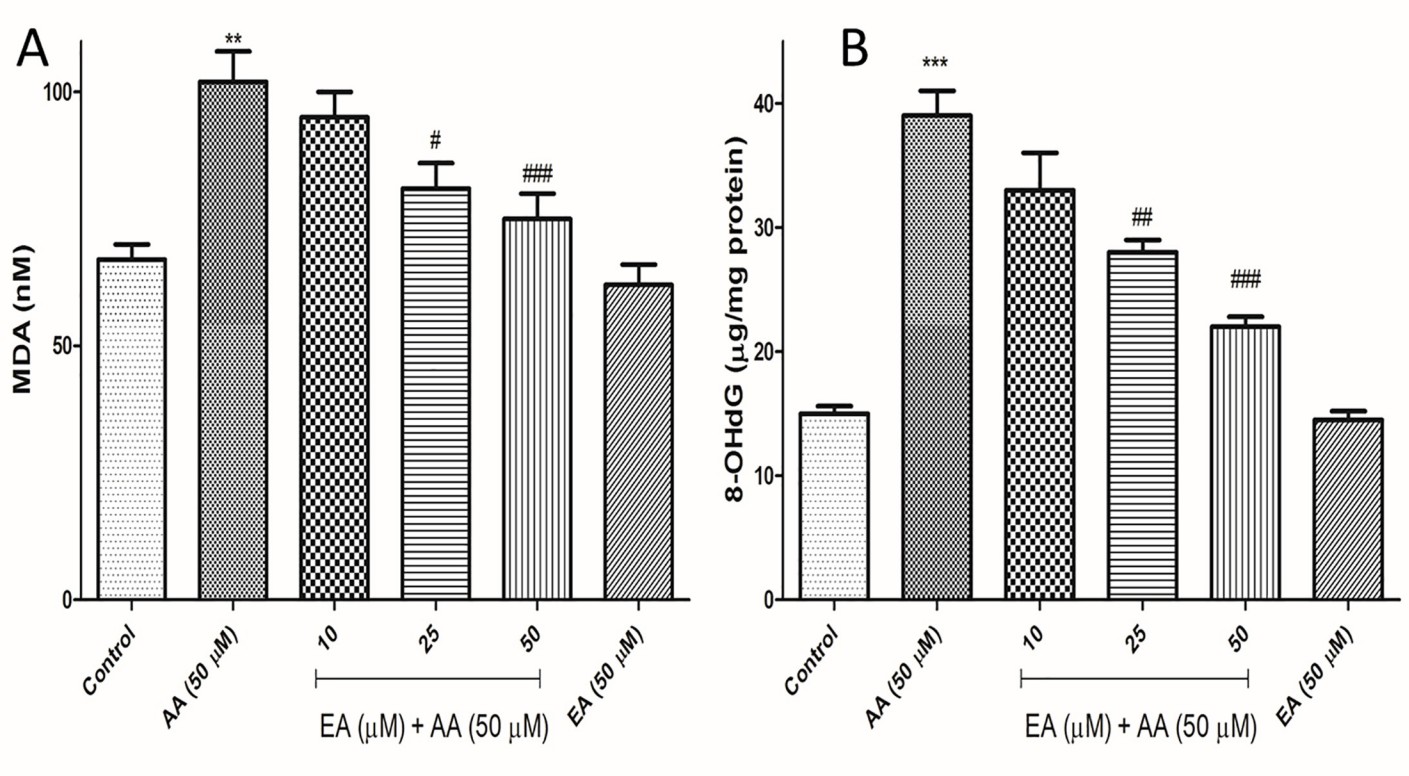

**Fig 6.** Addition of ellagic acid decreases AA-induced lipid peroxidation (A) and DNA oxidation (B) in human lymphocytes. (A) Representative measurements of lipid peroxidation following AA exposure (4 h) in the presence or absence of EA in human lymphocytes. Graph B presents measurements of DNA oxidation using kite following AA exposure (4 h) in the presence or absence of EA in human lymphocytes. Results are shown as mean ± SD, n = 3 technical replicates. All experiments were repeated at least three times. ***p < 0.001: Control versus AA; #p < 0.05, ###p < 0.01, ###p < 0.001: AA + EA versus AA-treated human lymphocytes ANOVA, Tukey's test.

antioxidant capable of preventing adverse effect of free radicals such as ROS and reactive nitrogen species (RNS). In this process, GSH oxidized to GSSG [38]. Our result showed acrylamide led to depletion of GSH level of human lymphocytes which is consistent with previously published studies. However, it has been reported that acrylamide without directly damaging DNA, can induce chromosome aberration and sister chromatid exchange [30].

Lysosomes and mitochondria are necessary for cellular metabolism and building blocks. Moreover, these organelles do other functions as fundamental signaling platforms in the cell that control many key processes such as proliferation, autophagy and cell death [39]. The interdependence of lysosomes and mitochondria is underscored by many lines of evidence obtained from cells [39]. Many lysosomal storage diseases have reports of mitochondrial dysfunction, reciprocally most perturbations of lysosomal function in mitochondrial diseases have been reported [39]. Loss of mitochondrial function is associated with increased ROS production, decreased mitochondrial ATP and mitochondrial fragmentation [40], each potentially affecting lysosomes. It has been reported that lysosomal function is rescued by antioxidant agents, suggesting that oxidative damages play a significant role in the lysosomal phenotype arising as an outcome of ROS mitochondrial. It has been reported that acrylamide induces mitochondrial dysfunction associated with ROS formation in several in vitro and in vivo studies [41]. Also presented data in the current work consistent with other published studies, showed mitochondrial and lysosomal dysfunction induced by acrylamide in isolated human

lymphocytes. Due to the oxidative stress, mitochondrial and lysosomal dysfunction are key feature of acrylamide cellular toxicity and also relation between these features together [40,42], therefore it is pivotal for cellular function to protect the mitochondria and lysosomes against oxidative damages.

The most important mechanisms that play a vital role during acrylamide-induced cytotoxicity and carcinogenicity are its metabolism to active intermediate glycidamide, mitochondrial dysfunction and ROS formation. Therefore, the use of dietary antioxidants such as EA, is very important to minimize the toxic effects associated with acrylamide. EA as an important component of vegetables and fruits and has been indicated to possess numerous antimutagenic and anticarcinogenic properties towards various carcinogenic agents [43]. Several studies have been designed to describe the broad anticarcinogenic and antimutagenic effects of EA [43]. The inhibition of CYP450 enzymes, especially CYP2E1 by EA is one of the proposed mechanisms [43]. CYP2E1 is important in the field of acrylamide-induced cytotoxicity and carcinogenicity [44]. It has been reported that daily intake of the EA decreases CYP2E1 protein level in the liver as a major enzyme in acrylamide activation [45]. Hence, EA administration might decrease acrylamide-induced cytotoxicity and carcinogenicity by inhibiting CYP2E1 and reduce reactive oxygen species by its antioxidant activity. Another mechanism for anticarcinogenic and antimutagenic effects of EA is its antioxidant potential. The antioxidant effect of EA is well known and this trait has been attributed to its free radical scavenging activity [46]. The existence of four hydroxyl functional groups and two lactone in EA chemical structure authorizes to scavenge a wide different of RNS and ROS [47]. Antioxidant activity of EA is not decreased after metabolism, its metabolites are also able of efficiently scavenging very free radicals, even faster than EA. This property of EA enhances antioxidant activity of EA at low concentrations, which is an desirable and unusual behavior for an common antioxidant [48]. Previous researches have been shown EA even at µM concentrations is effective in inhibiting lipid peroxidation [49]. Moreover, it has been reported the antioxidant properties of EA decrease mitochondrial dysfunction [50].

## Conclusion

Our result in the current study showed that ellagic acid significantly reduce cytotoxicity, mitochondrial and lysosomal damages, oxidative parameters and DNA oxidation in isolated lymphocytes. Neuroprotective and hepatotoxicity effects of EA against acrylamide are proved in animal model [16,17] and this study proved promising effects of EA in inhibiting oxidative stress, mitochondrial/lysosomal damages, cytotoxicity and DNA oxidation in human lymphocytes. As a limitation in the current study, the obtained findings must be verified by other experimental models such as animal studies, to establish the protective effects of ellagic acid in human exposed with acrylamide.

## Supporting information

**S1 File.**
(PZF)

**S2 File.**
(PZF)

**S3 File.**
(PZF)

**S4 File.**
(PZF)

**S5 File.**
(PZF)

**S6 File.**
(PZF)

**S7 File.**
(PZF)

**S8 File.**
(PZF)

**S9 File.**
(PZF)

## Acknowledgments

The data provided in this article was extracted from the Pharm D. thesis of Dr. Elahe Baghal. The thesis was conducted under supervision of Dr. Ahmad Salimi at Department of Pharmacology and Toxicology, School of Pharmacy, Ardabil University of Medical Sciences, Ardabil, Iran.

## Author Contributions

**Conceptualization:** Ahmad Salimi.

**Data curation:** Elahe Baghal, Niloufar Hashemidanesh, Farzad Khodaparast.

**Formal analysis:** Ahmad Salimi, Enayatollah Seydi.

**Funding acquisition:** Ahmad Salimi.

**Investigation:** Ahmad Salimi.

**Methodology:** Ahmad Salimi.

**Software:** Ahmad Salimi, Enayatollah Seydi.

**Supervision:** Ahmad Salimi.

**Writing – original draft:** Ahmad Salimi, Enayatollah Seydi.

**Writing – review & editing:** Ahmad Salimi, Hassan Ghobadi.

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
