## [Decision Letter · Decision Letter 0]

8 Jan 2021

PONE-D-20-36110

Mitochondrial, Lysosomal and DNA Damages Induced by Acrylamide Attenuate by Ellagic Acid in Human Lymphocyte

PLOS ONE

Dear Dr. Salimi,

Thank you for submitting your manuscript to PLOS ONE. After careful consideration, we feel that it has merit but does not fully meet PLOS ONE’s publication criteria as it currently stands. Therefore, we invite you to submit a revised version of the manuscript that addresses the points raised during the review process.

We have received the comments from our reviewers on your manuscript 'Mitochondrial, Lysosomal and DNA Damages Induced by Acrylamide Attenuate by Ellagic Acid in Human Lymphocyte' submitted to PLOS One. The comments of the reviewers are included at the bottom of this email.

Based on the reviewers comments I have decided that your manuscript can be accepted for publication after you have carried out the corrections as suggested by the reviewer(s). When preparing your revised manuscript, carefully consider the reviewer comments and provide a point-by-point response.

We look forward to receiving your revised manuscript.

Kind regards,

Soumen Bera, PhD

Academic Editor

PLOS ONE

Journal Requirements:

3. Thank you for stating in the text of your manuscript "Before blood sampling, the informed consent form was signed by the participants." Please also add this information to your ethics statement in the online submission form.

5. We noticed you have some minor occurrence of overlapping text with the following previous publications, which needs to be addressed:

- http://www.scielo.br/scielo.php?script=sci_arttext&pid=S1516-89132018000100426&lng=en&tlng=en

- https://doaj.org/article/5502805a216b4c0792fd0c32903148c5

- https://journals.sagepub.com/doi/10.1177/1091581820902405

In your revision ensure you cite all your sources (including your own works), and quote or rephrase any duplicated text outside the methods section. Further consideration is dependent on these concerns being addressed.

Reviewers' comments:

Reviewer's Responses to Questions

**Comments to the Author**

1. Is the manuscript technically sound, and do the data support the conclusions?

Reviewer #1: Yes

Reviewer #2: Yes

2. Has the statistical analysis been performed appropriately and rigorously? 

Reviewer #1: Yes

Reviewer #2: Yes

3. Have the authors made all data underlying the findings in their manuscript fully available?

Reviewer #1: Yes

Reviewer #2: Yes

4. Is the manuscript presented in an intelligible fashion and written in standard English?

Reviewer #1: No

Reviewer #2: Yes

5. Review Comments to the Author

Reviewer #1: The manuscript by Baghal et al. describes the consequences of the exposure to human lymphocytes to acrylamide and the protection offered to those cells by the simultaneous treatment with ellagic acid. Acrylamide toxicity has been extensively studied, but both the use of primary human lymphocytes and description of the protection offered by ellagic acid are novel results. Several different endpoints where considered and the data supports the conclusions that ellagic acid could prevent mitochondrial damage and oxidative stress due to acrylamide exposure. The data is presented concisely, and the experiments were performed carefully. However, the manuscript would benefit from editing to improve grammar and wording as well as correct typographical errors. Specific issues are presented below:

1. Several qualities of acrylamide are provided by the manufacturer and it should be specified which was obtained for the described experiments.

2. Some additional information regarding participants in the study, such as gender and how were the healthy individuals recruited, would be useful.

3. The specific software used for the reported analyses should be specified in the Methods section.

4. The oxidative lesion 8-OHdG often occurs during handling and storage. Where there any specific precautions taken to limit this from happening? Also, the production of 8-OHdG was not formally determined, just the steady state levels. It is possible that it was the removal of the lesion that was affected by the treatments.

5. The Discussion could be shortened to provide less general background and instead focus on the significance and implications of the data obtained.

6. In the References, reference 1 lacks details and the formatting of many references requires correction.

Reviewer #2: Comment

The work presented provides evidence that ellagic protects lymphocytes from the toxic effects of ellagic acid. The authors show that several end points for mitochondrial function and oxidative stress induced by acrylamide are inhibited by ellagic acid.

1. The author need to provide more physiological evidence for there findings in a cell or animal model to support the in vitro lymphocyte data presented.

2. Can the authors provided evidence for the mechanism of ellagic acid and how it blocks the toxicity of acrylamide?

3. Is the protection by ellagic acid related to its antioxidant properties or some other mechanism?

6. PLOS authors have the option to publish the peer review history of their article (what does this mean?). If published, this will include your full peer review and any attached files.

Reviewer #1: No

Reviewer #2: No

---

## [Author Response · Author response to Decision Letter 0]

5 Feb 2021

Prof. Soumen Bera, PhD

Academic Editor

PLOS ONE 

Dear Prof. Bera,

 Thank you very much for your letter of 8-Jan-2021, regarding the reviewers' comments on our submitted manuscript PONE-D-20-36110 entitled " Mitochondrial, Lysosomal and DNA Damages Induced by Acrylamide Attenuate by Ellagic Acid in Human Lymphocyte ". We carefully read the reviewers' comments and included their requests in our revised article and improved it according to their wish. I hope it is now satisfactory. In the following, please see our answers to the reviewers' comments point by point.

In the end, I would like to thank you and the respected reviewers for dedicating precious time and offering scientific comments.

 With kind regards 

 Ahmad Salimi

Comments to the Author

1. Is the manuscript technically sound, and do the data support the conclusions?

Reviewer #1: Yes

Reviewer #2: Yes

2. Has the statistical analysis been performed appropriately and rigorously?

Reviewer #1: Yes

Reviewer #2: Yes

3. Have the authors made all data underlying the findings in their manuscript fully available?

Reviewer #1: Yes

Reviewer #2: Yes

4. Is the manuscript presented in an intelligible fashion and written in standard English?

Reviewer #1: No

Reviewer #2: Yes

Our response: As requested by respected reviewer we checked the whole manuscript for typographical or grammatical errors and corrected them. Please refer to revised version and see yellow bands in the revised manuscript.

5. Review Comments to the Author

Reviewer #1: The manuscript by Baghal et al. describes the consequences of the exposure to human lymphocytes to acrylamide and the protection offered to those cells by the simultaneous treatment with ellagic acid. Acrylamide toxicity has been extensively studied, but both the use of primary human lymphocytes and description of the protection offered by ellagic acid are novel results. Several different endpoints where considered and the data supports the conclusions that ellagic acid could prevent mitochondrial damage and oxidative stress due to acrylamide exposure. The data is presented concisely, and the experiments were performed carefully. However, the manuscript would benefit from editing to improve grammar and wording as well as correct typographical errors. Specific issues are presented below:

1. Several qualities of acrylamide are provided by the manufacturer and it should be specified which was obtained for the described experiments.

Our response: As requested by respected reviewer we added the CAS number acrylamide in the materials and methods in the chemical and kits section. Please refer to revised version and see yellow bands in the revised manuscript.

2. Some additional information regarding participants in the study, such as gender and how were the healthy individuals recruited, would be useful.

Our response: As requested by respected reviewer we added additional information regarding participants in the study. Please refer to revised version and see yellow bands in the revised manuscript.

3. The specific software used for the reported analyses should be specified in the Methods section.

Our response: As requested by respected reviewer we added the necessary information for statistical software and other information about statistical analysis methods in the text. Please refer to revised version and see yellow bands in the revised manuscript.

4. The oxidative lesion 8-OHdG often occurs during handling and storage. Where there any specific precautions taken to limit this from happening? Also, the production of 8-OHdG was not formally determined, just the steady state levels. It is possible that it was the removal of the lesion that was affected by the treatments.

Our response: Thank you very much for your good question, in the current study we measured 8-OHdG as DNA Damage indicator exactly after 4 hours of exposure and compared with untreated control for remove of other interventions. Also, we used the standard protocol for detection of 8-OHdG according previous studies. 

5. The Discussion could be shortened to provide less general background and instead focus on the significance and implications of the data obtained.

Our response: Thank you very much for your good comment, as requested by respected reviewer we shorted the Discussion and focus on the significance and implications of the data obtained as much as possible and added the limitation of the study. Please refer to revised version and see yellow bands in the revised manuscript.

6. In the References, reference 1 lacks details and the formatting of many references requires correction.

Our response: Thank you very much for your good comment, as requested by respected reviewer we corrected the reference 1. Please refer to revised version and see yellow bands in the revised manuscript.

Reviewer #2: Comment

The work presented provides evidence that ellagic protects lymphocytes from the toxic effects of ellagic acid. The authors show that several end points for mitochondrial function and oxidative stress induced by acrylamide are inhibited by ellagic acid.

1. The author need to provide more physiological evidence for there findings in a cell or animal model to support the in vitro lymphocyte data presented.

Our response: Thank you very much for your comment as you know we used the isolated human lymphocytes in this study as wild type cells and useful model for assessment and evaluation of toxicity induced by drug and chemicals, in here we showed that ellagic acid with antioxidant potential reduced the toxic effect of ellagic acid. We used BHT as a known antioxidant for confirmation (data not shown in the graphs). Due to the fact that this study was performed on primary human lymphocytes, it is not suitable for animal models. As main limitation in the current study we suggested in the last paragraph in the revised manuscript. Please refer to revised version and see yellow bands in the revised manuscript. 

2. Can the authors provided evidence for the mechanism of ellagic acid and how it blocks the toxicity of acrylamide?

Our response: In the current study we used the butylated hydroxytoluene (BTH) as a known antioxidant for confirmation of this fat that ellagic acid through antioxidant potential reduce the acrylamide-induced toxicity in human lymphocytes. As requested by respected reviewer we added the details in the revised version. Please refer to revised version and see yellow bands in the revised manuscript. 

3. Is the protection by ellagic acid related to its antioxidant properties or some other mechanism?

Our response: Yes. Please see above our responses (In the current study we used the butylated hydroxytoluene (BTH) as a known antioxidant for confirmation of this fat that ellagic acid through antioxidant potential reduce the acrylamide-induced toxicity in human lymphocytes).

6. PLOS authors have the option to publish the peer review history of their article (what does this mean?). If published, this will include your full peer review and any attached files.

Do you want your identity to be public for this peer review? For information about this choice, including consent withdrawal, please see our Privacy Policy.

Reviewer #1: No

Reviewer #2: No

---

## [Editor Report · Decision Letter 1]

15 Feb 2021

Mitochondrial, Lysosomal and DNA Damages Induced by Acrylamide Attenuate by Ellagic Acid in Human Lymphocyte

PONE-D-20-36110R1

Dear Dr. Salimi,

We’re pleased to inform you that your manuscript has been judged scientifically suitable for publication and will be formally accepted for publication once it meets all outstanding technical requirements.

Kind regards,

Soumen Bera, PhD

Academic Editor

PLOS ONE

---

## [Editor Report · Acceptance letter]

19 Feb 2021

PONE-D-20-36110R1 

Mitochondrial, Lysosomal and DNA Damages Induced by Acrylamide Attenuate by Ellagic Acid in Human Lymphocyte 

Dear Dr. Salimi:

I'm pleased to inform you that your manuscript has been deemed suitable for publication in PLOS ONE. Congratulations! Your manuscript is now with our production department. 

Kind regards, 

on behalf of

Dr. Soumen Bera 

Academic Editor

PLOS ONE